# Increased Number of Mucosal-Associated Invariant T Cells Is Associated with the Inhibition of Nonalcoholic Fatty Liver Disease in High Fat Diet–Fed Mice

**DOI:** 10.3390/ijms232315309

**Published:** 2022-12-04

**Authors:** Haruka Kishi, Isao Usui, Teruo Jojima, Shiho Fujisaka, Sho Wakamatsu, Yuiko Mizunuma-Inoue, Takafumi Niitani, Shintaro Sakurai, Toshie Iijima, Takuya Tomaru, Kazuyuki Tobe, Yoshimasa Aso

**Affiliations:** 1Department of Endocrinology and Metabolism, Dokkyo Medical University, Tochigi 321-0293, Japan; 2First Department of Internal Medicine, University of Toyama, Toyama 930-0194, Japan

**Keywords:** MAIT cells, NAFLD, lipogenesis, innate-like T lymphocyte, dyslipidemia

## Abstract

Nonalcoholic fatty liver disease (NAFLD) is an emerging worldwide health concern. The disease may involve immune cells including T cells, but little is known about the role(s) of the innate-like T cells in the liver. Furthermore, the most abundant innate-like T cells in the human liver are mucosal-associated invariant T (MAIT) cells, but the involvement of MAIT cells in NAFLD remains largely unexplored because of their paucity in mice. In this study, we used a novel mouse line, Vα19, in which the number of MAIT cells is equivalent to or greater than that in humans. Compared with the control mice, Vα19 mice fed a high-fat diet (HFD) exhibited a reduction in lipid accumulation, NAFLD activity score, and transcripts relevant to lipogenesis. In addition, serum triglyceride and non-esterified fatty acids were lower in Vα19 mice fed normal chow or HFD. In contrast, the Vα19 mice showed little or no change in glucose tolerance, insulin sensitivity, inflammation in adipose tissues, or intestinal permeability compared with the controls, irrespective of diet. These results suggest that the presence of MAIT cells is associated with reduced lipogenesis and lipid accumulation in the liver; however, further studies are needed to clarify the role of MAIT cells in hepatic lipid metabolism.

## 1. Introduction

Nonalcoholic fatty liver disease (NAFLD) is a pathological liver condition diagnosed by a histological or imaging examination, excluding other liver diseases. Because the prevalence of NAFLD is increasing worldwide and the disease is becoming a major public health concern, preventive and treatment strategies are needed, and research aims to develop them through elucidating the pathogenesis of the disease. Most cases of NAFLD develop on the basis of or in conjunction with obesity, insulin resistance, type 2 diabetes, or dyslipidemia. In this sense, NAFLD is sometimes referred to as metabolic dysfunction–associated fatty liver disease (MAFLD) [1,2]. The “two-hit theory” has long been known as a classical pathogenesis model of NAFLD. In this model, the first hit comprises metabolic disturbance, which increases lipid accumulation in the liver, resulting in fatty liver. Then, as the second hit, various stimuli such as oxidative stress induce inflammation and fibrosis in the fatty liver, eventually leading to the development of hepatocellular carcinoma [3,4]. In addition, the progression of NAFLD from the early stages is also influenced by changes in gut microbiota associated with metabolic disturbances [5]. The histology of NAFLD is characterized by steatosis, lobular inflammation, and ballooning of hepatocytes. Neutrophils are primarily responsible for the lobular inflammation, which is followed by infiltration of multiple immune cells, such as macrophages and lymphocytes, as the NAFLD progresses. These immune cells have been reported to play important roles in the pathogenesis of NAFLD [6], but the involvement of unconventional innate-like T cells is not fully understood.

Mucosal-associated invariant T (MAIT) cells belong to the family of innate-like T cells that includes natural killer T and γδT cells. Innate-like T cells form a distinctive T-cell fraction because they have innate immune cell–like characteristics such as cytotoxic effector function [7,8,9]. They express the T cell receptor (TCR), but the receptor is different from the conventional TCR. For example, the semi-invariant TCRa of MAIT cells has a restricted combination of variable region (V) and joint region (J) of Vα19-Jα33 (mouse) and Vα7.2-Jα33 (human), which is paired with a limited set of TCR β repertoire [10,11,12,13]. The MAIT cell TCR recognizes small compounds as specific ligands, such as bacterial vitamin B2 and B9 metabolites and their derivatives, which are presented on major histocompatibility complex (MHC) I–related gene protein (MR1). Furthermore, MAIT cells also express interleukin 12 (IL-12)/IL-18 receptors independently of TCR, allowing the cytokines IL-12 and IL-18 to activate the MAIT cells.

Activated MAIT cells exhibit T helper 1 (Th1)/Th17 secretion patterns, such as interferon gamma (IFNγ), tumor necrosis factor alpha (TNF-α), and IL-17, and may be involved in the pathogenesis of various inflammatory diseases [14,15] and metabolic diseases [14,16,17,18,19]. In the physiological condition, the liver acts as a secondary barrier to pathogens from the intestinal tract, and hepatic immune cells such as Kupffer, natural killer, natural killer T, and MAIT cells play protective roles. Hepatic MAIT cells are continuously activated and participate in the activation of other immune cells [20,21]. Recently, studies found that in the late stages of NAFLD, MAIT cells promote liver fibrosis by secreting IL-17 [22,23]. On the other hand, another group reported that MAIT cells improve the inflammatory phenotype of NAFLD by regulating macrophage polarization [24]. These results suggest that the role of MAIT cells may differ depending on the stage of NAFLD. In addition, the roles of MAIT cells in the earlier stages of the disease, such as lipid accumulation, have not been elucidated.

MAIT cells are the most numerous T cells in humans; for example, they represent 20 to 45% of T cells in the liver [25]. On the other hand, the number of MAIT cells is very small in experimental animals, including mice [10,26]. Thus, to date, animal models have not been able to be used to fully examine the precise role of MAIT cells in diseases, including NAFLD. To address this issue, Sugimoto et al. recently established induced pluripotent stem cells from murine MAIT cells (MAIT-iPSCs) [27] and then generated a novel mouse model by using MAIT-iPSCs. These mice, designated as Vα19 mice, express a rearranged TCRa locus that is specific to MAIT-cell TCRs. They also express a higher number of MAIT cells, which can be activated by TCR agonists and cytokines to produce IFN-γ and IL-17 [28]. The use of these mice has enabled investigations of the effects of increased MAIT cell numbers on the pathogenesis of various diseases.

The present study was performed to clarify the roles of MAIT cells in the pathogenesis of NAFLD and other metabolic disorders associated with obesity by examining Vα19 mice fed a high-fat diet (HFD). We demonstrated that the increased number of MAIT cells was associated with a suppression of lipogenesis in the liver and an improvement of NAFLD.

## 2. Results

To clarify the effects of increased numbers of MAIT cells on the pathogenesis of NAFLD and other systemic metabolic abnormalities due to diet-induced obesity, we performed experiments with Vα19 mice, which have abundant MAIT cells from birth. C57BL/6NJcl mice (hereinafter B6 mice) were used as the control. Both types of mice were fed normal chow (NC) and a HFD from 6 to 27 weeks of age (Appendix A). During this period, the HFD increased the body weight of both types of mice more strongly than the NC did (Figure 1a). There were no significant differences between the Vα19 mice and the B6 mice in body weight (Figure 1a) or food intake (Figure 1b). At 27 weeks of age, the effect of the increased number of MAIT cells on liver function was investigated. MAIT cells in the liver were identified by flow cytometry as CD45-positive/TCRβ-positive/MR1-tetramer-positive/Ac 6FP-negative cells (Appendix A). Hepatic MAIT cell numbers were almost 50 to 100 times higher in the Vα19 mice than in the B6 mice. The HFD did not affect the number of MAIT cells significantly in the livers of the Vα19 mice (Appendix A and Figure 1c). Liver weight was not significantly different between the Vα19 and B6 mice on NC and HFD (Figure 1d).

The activity of NAFLD was semi-quantitatively evaluated by the NAFLD activity score (NAS) in liver tissues stained with hematoxylin and eosin, as previously described [29]. Because the NC-fed mice showed almost no pathological changes as steatohepatitis, the NAS and its components, i.e., steatosis, lobular inflammation, and hepatocyte ballooning were compared only between the Vα19 and B6 mice fed the HFD (Figure 2a–c). NAS was significantly lower in the Vα19 mice than in the B6 mice (Figure 2a,b). When the NAS components were compared, only steatosis significantly decreased in the Vα19 mice, and no significant differences were seen in lobular inflammation or hepatocyte ballooning between the Vα19 and B6 mice (Figure 2c).

To confirm the results of the NAS in the Vα19 and B6 mice, we performed several additional experiments. We assessed lipid accumulation in the liver by direct quantification of triglyceride content (Figure 2d) and Sudan III staining (Figure 2e). Both assessments confirmed that triglyceride accumulation in the liver was significantly lower in the Vα19 mice than in the B6 mice (Figure 2d,e). Then we evaluated hepatic inflammation by measuring the expression of inflammation-related genes in the liver. mRNA expressions of inflammatory cytokines, such as TNF-α and IL-6, and of the inflammatory M1 macrophage marker CD11c were upregulated by the HFD, but there were no significant differences between the Vα19 and B6 mice (Figure 2f). In addition, we assessed hepatocyte degeneration by measuring serum aspartate transaminase (AST) and alanine transaminase (ALT) levels, but neither was significantly different between the Vα19 and B6 mice. These results suggest that an increase in MAIT cell number is associated with a reduction in lipid accumulation but does not affect inflammation or hepatocyte degeneration in HFD-fed mice.

Next, we investigated the mechanism by which lipid accumulation in the livers of the HFD-fed Vα19 mice reduced compared with the HFD-fed B6 mice. The lipid content of the liver is regulated by lipogenesis and/or lipid catabolism, including β-oxidation. Our quantitative real-time polymerase chain reaction (RT-PCR) experiments demonstrated that the expressions of some genes involved in the lipogenic process, such as SREBP-1c, ACC1, and FAS, were significantly downregulated in the Vα19 mice (Figure 3a). On the other hand, there was no significant difference between the Vα19 and B6 mice in β-oxidation–related genes such as CPT1A, Acox1/2, and PPARα (Figure 3b). These results suggest that the reduced lipid accumulation observed in the Vα19 mice is not due to increased β-oxidation but to decreased lipid synthesis in the liver. To further study the effects of reduction in hepatic lipogenesis, we compared serum lipid profiles between the Vα19 and B6 mice. The serum triglyceride level of the Vα19 mice was significantly lower than that of the B6 mice only in the HFD-fed mice, but the non-esterified fatty acid (NEFA) level was lower in both the NC- and HFD-fed Vα19 mice. No significant differences were observed in the total cholesterol (Figure 4).

Subsequently, we examined the association between the increased number of MAIT cells and the glucose metabolism. In the HFD-fed mice, the liver expression of phosphoenolpyruvate carboxykinase (PEPCK), one of the rate-limiting enzymes of hepatic gluconeogenesis, was significantly lower in the Vα19 than in the B6 mice. Similarly, peroxisome proliferator–activated receptor gamma coactivator 1-alpha (PGC-1a), a transcriptional coactivator regulating the transcription of some gluconeogenetic enzymes, including PEPCK, also decreased in the Vα19 mice (Figure 5a). However, when glucose metabolism and insulin sensitivity were evaluated by an intraperitoneal glucose tolerance test and insulin tolerance test, respectively, no significant differences were found between the NC- and HFD-fed Vα19 and B6 mice (Figure 5b,c). These results suggest that although the reduced hepatic lipid accumulation was associated with the downregulation of gluconeogenesis, the impact of these changes in the liver was not strong enough to affect systemic glucose metabolism and insulin resistance. In addition, to clarify the involvement of the activation of TCR signaling of MAIT cells on the glucose metabolism, we further performed an intraperitoneal glucose tolerance test and an insulin tolerance test after administration of 5-OP-RU, a specific ligand for MAIT-cell TCRs. The administration of the 5-OP-RU did not affect glucose levels in the glucose tolerance and insulin tolerance tests in the NC-fed Vα19 mice, suggesting that activation of TCR signaling in MAIT cells does not have an impact on the regulation of systemic glucose tolerance and insulin resistance.

The inflammatory remodeling of visceral adipose tissue is known to play an important role in the pathogenesis of obesity-related metabolic abnormalities such as insulin resistance. Therefore, we next investigated the effect of increased MAIT cell number on HFD-induced remodeling of epididymal white adipose tissue (eWAT). The Vα19 mice had 4 to 10 times more MAIT cells in eWAT than the B6 mice did. The HFD did not affect the number of MAIT cells in eWAT in either the Vα19 or the B6 mice. Moreover, the eWAT weight and adipocyte size increased in the HFD-fed mice but were not significantly different between the Vα19 and B6 mice on the HFD and NC (Figure 6a,b). Consistent with these results, no significant differences were observed between the Vα19 and B6 mice in the expressions in eWAT of preadipocyte-marker genes, such as Pref-1 and PDGFRa (Figure 6c); macrophage-related genes, such as F4/80; and inflammation-related genes, such as TNF-α and IL-6 (Figure 6d,e). These results suggest that the higher number of MAIT cells in Vα19 mice does not affect the HFD-induced pathological remodeling of eWAT.

Among all the tissues in mice, the intestinal mucosa has the highest concentration of MAIT cells. Abnormalities in intestinal function observed in obesity are known to adversely affect the systemic metabolism. Consequently, we investigated the effect of MAIT cell number on intestinal permeability by measuring the leakage of orally administered fluorescein isothiocyanate (FITC)-dextran into the blood. We found no significant difference in the concentration of FITC-dextran in circulating blood between the Vα19 and B6 mice fed both HFD and NC (Appendix A), suggesting that the increased number of MAIT cells in Vα19 mice does not affect intestinal permeability.

## 3. Discussion

In this study, we examined the association of MAIT cells with metabolic features of HFD-induced obesity using Vα19 mice, a recently generated mouse strain that has a higher number of MAIT cells than other mice. Compared with the B6 control mice, the Vα19 mice fed a HFD diet had a lower NAS, reduced lipid accumulation, downregulated lipogenic gene expression in the liver, and reduced serum triglyceride and NEFA levels. These results suggest that a higher number of MAIT cells is associated with a decrease in diet-induced lipogenesis and lipid accumulation in the liver.

Recently, several studies demonstrated the physiological and pathological functions of MAIT cells in the liver. For example, regarding physiological condition, hepatic MAIT cells play protective roles against pathogens that cross the immune barrier of the intestinal tract [20,21,30]. On the other hand, in the late stages of NAFLD, MAIT cells are reported to promote liver fibrosis by secreting IL-17 and activating stellate cells [22,23]. In this study, we did not evaluate the fibrotic changes in the liver because the HFD feeding did not induce liver fibrosis significantly. To confirm the role of MAIT cells in the fibrotic process, experiments can be carried out using Vα19 mice fed a diet containing high fat, high carbohydrate, and cholesterol, which is known to induce fibrosis more strongly than HFD. We would like to address this theme in the next study.

Li et al. recently reported interesting data on the relation between the number of MAIT cells and NAFLD. They found that the number of MAIT cells in the liver of NAFLD patients was higher than in healthy controls and was positively correlated with NAS [24]. The number of MAIT cells was also higher in the liver of an NAFLD model mouse created by feeding a methionine-choline–deficient diet (MCD). Because MR1-deficient mice, which are completely deficient in MAIT cells, exhibited an increase in the ratio of liver weight to body weight and a higher NAS on MCD feeding, Li et al. suggested that MAIT cells play a protective (i.e., suppressive) role in the pathogenesis of NAFLD [24]. Their data are consistent with our present results, but they did not show that the increase in MAIT cell number directly suppressed NAFLD. In the present study, we analyzed Vα19 mice, a recently generated mouse strain that is rich in MAIT cells. By analyzing these mice, we demonstrated that the increase in MAIT cell number was associated with reduction in hepatic lipogenesis and NAS (Figure 2 and Figure 3a). The Vα19 mouse has been generated recently, and the phenotype of this novel mouse model has not been fully evaluated, including the changes in inflammatory/immune cells in the liver. Thus, it is also possible that the reduction in steatosis of Vα19 mice was due to changes in the other inflammatory or immune cells. These results are important in that they added new evidence supporting an association between increased MAIT cell number and NAFLD.

The study by Li et al. also demonstrated that MAIT cells exert anti-inflammatory effects by inducing M2 polarity of intrahepatic macrophages through increased production of IL-4. In contrast, we did not find any significant changes in inflammation-related markers in the liver of Vα19 mice (Figure 2a,c,f). The reason for the different results regarding the anti-inflammatory effects of MAIT cells is unclear; however, because Li et al. observed the anti-inflammatory response only in MCD-fed mice, the differences may be due to the methods used to induce NAFLD, i.e., MCD or HFD. Further analyses are needed to assess these possibilities.

Another important finding of this study was that almost no differences were observed in glucose tolerance, insulin sensitivity, inflammatory changes in adipose tissues, or intestinal permeability between the Vα19 and control mice fed NC and HFD. These results suggest that an increase in MAIT cells alone does not affect these metabolic parameters, which are associated with diet-induced obesity. These results are different from a previous study by Toubal et al., which showed that MAIT cells were associated with the worsening of these metabolic parameters [19]. The reasons for the different effects of MAIT cell number on these metabolic markers in our study and the previous study are not known, but they may be due to the use of different mouse models rich in MAIT cells: Toubal et al. created their mouse model by using a transgenic technique to overexpress Vα19, whereas the Vα19 mice in our study were created with iPS cells established from mouse MAIT cells. The iPS technique has the advantage that expression of Vα19 is regulated by the endogenous promoter. This important issue will be addressed in our next study.

The most important limitation of this study was that we could not demonstrate the mechanism by which the increased number of MAIT cells inhibited lipid synthesis and triglyceride accumulation in the liver. We hypothesize that some humoral factors secreted by MAIT cells may affect enzymatic activities critical for the process of lipid synthesis in hepatocytes. To detect humoral factors, we are currently planning co-culture experiments on MAIT cells and hepatocytes. By performing some OMICS analyses of culture media, we aim to detect such humoral factors in the next study. In addition, the lowering in NEFA possibly reduced triglyceride accumulation in the livers of the Vα19 mice. However, the causal relationship between the lowered NEFA and the reduced hepatic triglyceride accumulation has not been clarified. Further experiments are needed to address this possibility.

In conclusion, this study suggests that the presence of MAIT cells is associated with reduced lipogenesis and lipid accumulation in the liver; however, further studies are needed to clarify the role of MAIT cells in hepatic lipid metabolism.

## 4. Materials and Methods

### 4.1. Mice

Dr. Sugimoto, Prof. Wakao, and their research group (at the Host Defense Division, Research Center for Advanced Medical Science, Dokkyo Medical University, Tochigi, Japan) generated a model mouse rich in MAIT cells (Vα19 mouse) by using iPS technology, as described previously [27,28]. They kindly provided the Vα19 mice for our present study.

The Vα19 mice and their littermate controls (B6) were housed at the Animal Research Center, Dokkyo Medical University, with controlled temperature and lighting in a 12 h light–dark cycle. Only male mice were used for the experiments. The mice were randomly assigned to four groups, i.e., Vα19 mice fed HFD (MAIT-HFD), Vα19 mice fed NC (MAIT-NC), B6 mice fed HFD (B6-HFD), and B6 mice fed NC (B6-NC). The mice in the NC groups were fed a chow diet (CE-2; CLEA-Japan, Tokyo, Japan) and water ad libitum from weaning at 4 weeks until 27 weeks of age. The mice in the HFD groups were fed a chow diet from 4 to 6 weeks of age and then a HFD with 60% calories from fat (D12492, RESEACH DIETS, New Brunswick, NJ, USA) to 27 weeks of age. Most of the analyses were conducted at 27 weeks of age (Appendix A).

### 4.2. Flow Cytometry

Flow cytometry was performed as described previously [27,28]. Briefly, cells were incubated with antibodies against surface antigens (anti-CD45, anti-CD25, anti-TCRβ, anti-MR1tetramer, anti-CD8, anti-CD69, and anti-CD4) and flow cytometry block. After the final staining with 7-amino-actinomycin D to discriminate between live and dead cells, the cells were analyzed with the MACS Quant cell analyzer (3 lasers, 10 parameters; Miltenyi Biotec, Bergisch Gladbach, Germany) or the AttuneNxT acoustic focusing cytometer (4 lasers, 14 parameters; Thermo Fisher Scientific, Waltham, MA, USA). The data were processed with FlowJo software (version 9; BD Biosciences, Fremont, CA, USA). MAIT cells in the liver were identified as CD45-positive/TCRβ-positive/MR1-tetramer-positive/Ac 6FP-negative cells (Appendix A). Mouse MR1 5-OP-RU PE–labeled tetramer and mouse MR1 6-FP (control) PE–labeled tetramer were provided by the NIH Tetramer Core Facility (Emory University, Atlanta, GA, USA).

### 4.3. Histological Analysis

Liver samples were fixed in 4% neutral buffered formalin and embedded in paraffin. Tissues were sectioned (in 5 mm-thick sections) and routinely stained with hematoxylin and eosin (HE) with Lillie-Mayer hematoxylin (Muto Pure Chemicals, Tokyo, Japan) or Sudan III. Based on observation of the HE-stained samples, the NAS was calculated from the severity scores of steatosis, lobular inflammation, and hepatocyte ballooning according to the criteria of Kleiner et al. [31,32]. Epididymal adipose tissues were fixed in 10% neutral buffered formalin and embedded in paraffin. The tissues were sectioned (in 5 mm-thick sections) and stained with HE. A morphometric analysis was performed with an image analysis system (BZ-Analyzer; Keyence, Osaka, Japan). Adipocyte size was determined by capturing bright field images and measuring 500 to 800 cells per mouse.

### 4.4. Quantitative RT-PCR

The total RNA of liver and epididymal adipose tissue was extracted with the SV Total RNA Isolation System (Promega, Madison, WI, USA), and the cDNA was synthesized with SuperScript III First Strand Synthesis Super mix (Invitrogen, Carlsbad, CA, USA). Quantitative RT-PCR was performed using the TaqMan method (50 °C for 2 min, 95 °C for 10 min, and 40 cycles of 95 °C for 15 s and 60 °C for 1 min) with THUNDERBIRD^®^ probe quantitative PCR Mix (TOYOBO, Osaka, Japan) and the following premade primer sets (Applied Biosystems, Foster City, CA, USA): F4/80 (Mm00802529_m1), CD11c (Mm00498707_m1), CD206 (Mm01329362_m1), TNF-α (Mm00443260_g1), IL-6 (Mm00446190_m1), SREBP1c (Mm00550338_m1), ACC1 (Mm01304257_m1), ACC2 (Mm01204659_m1), FAS (Mm00662319_m1), PPARg (Mm00440940_m1), CPT1A (Mm01231183_m1), Acox1 (Mm00443579_m1), Acox2 (Mm00446408_m1), PPARα (Mm00440939_m1), PEPCK (Mm01247058_m1), G6PC (Mm00839363_m1), PGC1-a (Mm01208835_m1), Pref-1 (Mm00494477_m1), Sca-1 (Mm00726565_s1), PDGFRa (Mm00440701_m1), MCP1 (Mm00441242_m1), and iNOS (Mm00440502_m1). The relative abundance of transcripts was normalized according to the expression of 36B4 (Mm00725448_s1). The results were shown by the delta–delta Ct method to quantify the RT-PCR data.

### 4.5. Chemical Analysis of Blood and Liver Samples

Blood was collected from the tail vein and centrifuged at 1200 rpm for 15 min. The serum was stored at −80 °C. AST, ALT, total cholesterol, triglyceride, and non-esterified fatty acid were analyzed by enzymatic methods with an L-type Wako kit (ORIENTAL YEAST, Tokyo, Japan). For the measurement of triglyceride in the liver, 100 mg of liver samples were surgically harvested and washed with saline. The tissues were stored in liquid nitrogen until analysis. Triglyceride was extracted from the samples by the FOLCH method [33]. The amount of triglyceride was then measured by the enzymatic method.

### 4.6. Intraperitoneal Glucose Tolerance Test and Insulin Tolerance Test

Intraperitoneal glucose tolerance and intraperitoneal insulin tolerance tests were performed as reported previously [34]. Briefly, for the glucose tolerance test, the mice were injected with glucose (2 mg/g body weight) intraperitoneally, and for the insulin tolerance test, with human insulin (1.0 and 1.5 mU/g body weight for the NC- and HFD-fed mice, respectively). Blood samples were collected from the tail vein before and 15, 30, 60, 90, and 120 min after the glucose or insulin injection. Blood glucose levels were measured with the StatStrip (NIPRO, Tokyo, Japan), and serum insulin levels with a mouse insulin enzyme–linked immunosorbent assay (ELISA) kit (Morinaga Ultra-Sensitive Mouse/Rat Insulin ELISA Kit; MORINAGA, Yokohama, Japan). To evaluate the effects of activation of TCR signaling on glucose tolerance, 5-OP-RU, a specific ligand for the MAIT cell TCR, was injected 24 h before the insulin tolerance test. The 5-OP-RU was prepared by incubating an aliquot of 3.6 mM 5-A-RU in DMSO with three volumes of 1 mM methylglyoxal in water at 37 °C for 30 min.

### 4.7. Evaluation of Intestinal Permeability

Intestinal permeability was evaluated as described previously [35]. Briefly, FITC-dextran (Sigma-Aldrich, St. Louis, MO, USA) was administered orally, and the blood samples were collected by cardiac puncture 4 h after administration. The serum was diluted 1:1 (vol/vol) in phosphate-buffered serum, and the FITC-fluorescence intensity of each sample was measured with a fluorescence spectrometer (excitation, 485 nm; emission, 528 nm) The concentration of the FITC-dextran was calculated with a standard curve.

### 4.8. Statistical Analysis

Statistical analysis was performed with Prism (GraphPad Software, Version 8.4.1, San Diego, CA, USA). All the results were evaluated by an unpaired Student’s *t* test. The data was presented as mean ± SD, and differences with a *p* value of less than 0.05 were considered statistically significant.

## Figures and Tables

**Figure 1 ijms-23-15309-f001:**
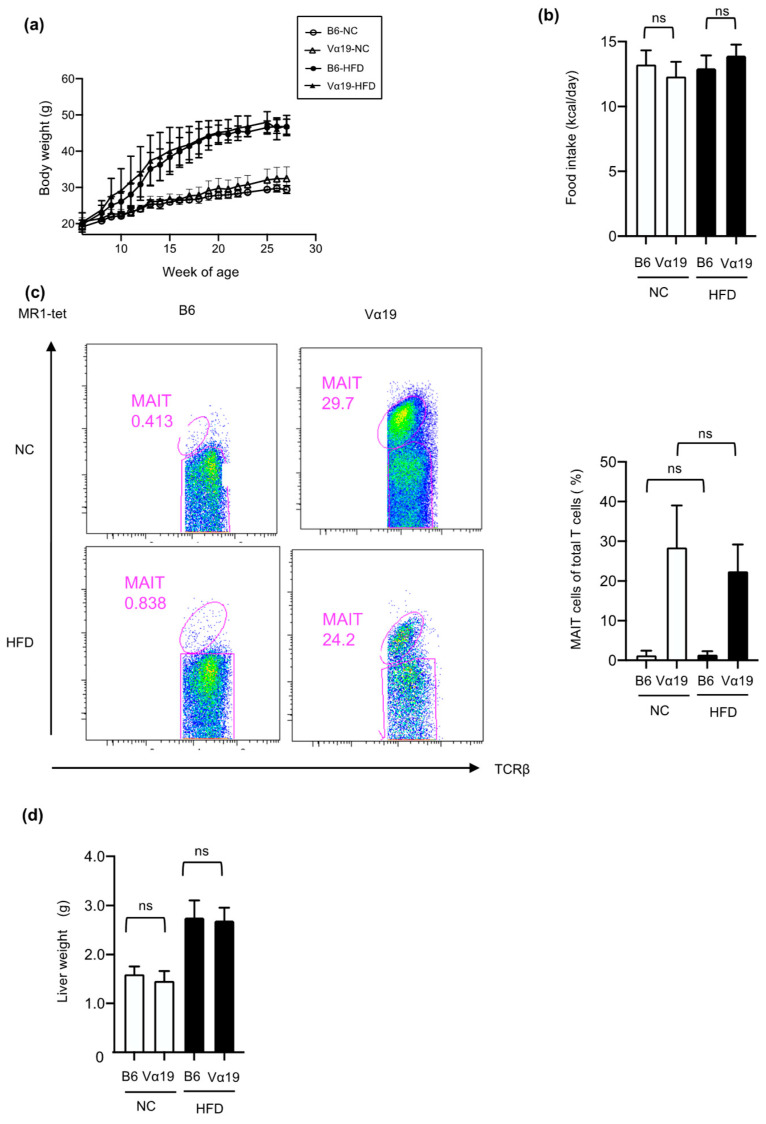
Body weight, food intake and liver weight were not different between Vα19 mice and C57BL/6 mice fed normal chow and a high-fat diet. (**a**) Development of body weight in Vα19 mice (Vα19) and C57BL/6 mice (B6). Open circles, B6 mice on normal chow (NC); open triangles, Vα19 mice on NC; closed circles, B6 mice on high-fat diet (HFD); closed triangles, Vα19 mice on HFD. (**b**) Food intake of Vα19 mice and B6 mice at age 24 to 27 weeks. (**c**) Flowcytometry analysis of liver cells from Vα19 and B6 mice fed NC and HFD. Cells in the fraction positive for both T cell receptor β and MR1-tetramer (MR1-tet) were defined as mucosal-associated invariant T (MAIT) cells. Representative results are shown. (**d**) Liver weight of Vα19 and B6 mice at 27 weeks of age. The results are shown as the mean ± SD for 6 to 12 mice per group. ns, not statistically significant.

**Figure 2 ijms-23-15309-f002:**
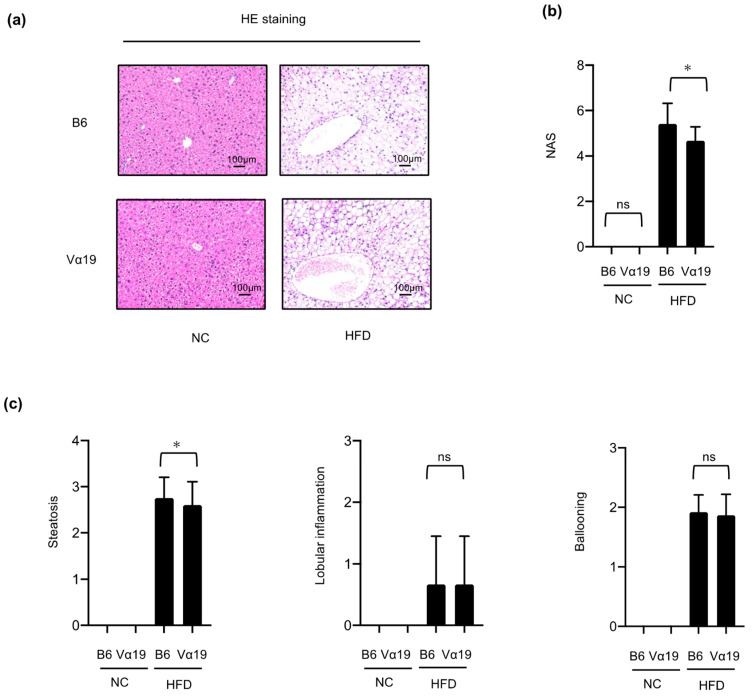
Non-alcoholic fatty liver disease activity score and lipid accumulation in the liver decreased in Vα19 mice fed a high-fat diet. (**a**) Liver sections stained with hematoxylin and eosin (HE) from Vα19 mice (Vα19) and C57BL/6 mice (B6) fed normal chow (NC) and a high-fat diet (HFD) at 27 weeks of age. Representative images are shown. Scale bar:100 μm. (**b**,**c**) Non-alcoholic fatty liver disease activity score (NAS) (**b**) and its components (**c**) were determined by examining 18 liver sections stained with HE. (**d**) Quantification of hepatic triglycerides. (**e**) Liver sections stained with Sudan III. (**f**) mRNA levels of macrophage- and inflammation-related genes. Data on each mRNA were normalized according to the 36B4 mRNA level. Results are shown as the mean ± SD for 4 to 6 mice per group. * *p* < 0.05, ns, not statistically significant.

**Figure 3 ijms-23-15309-f003:**
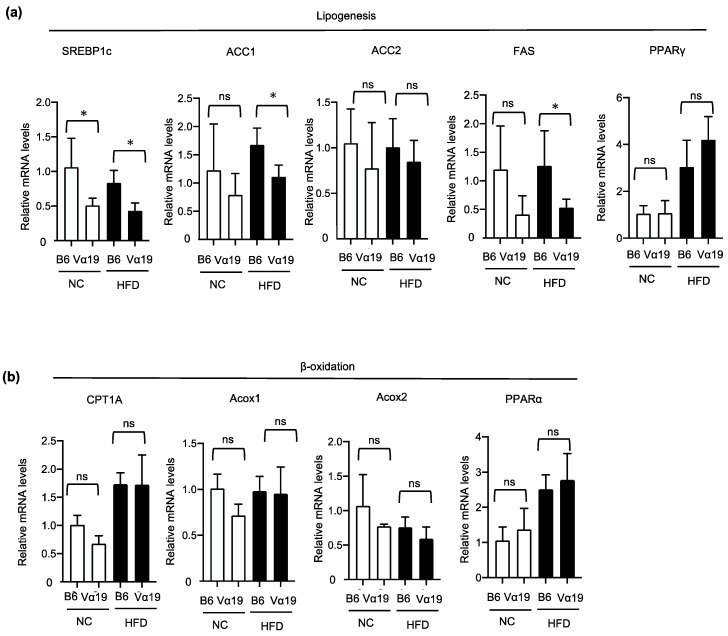
Lipogenesis-related genes were decreased in Vα19 mice fed a high-fat diet. (**a**,**b**) mRNA levels for lipogenesis- (**a**) or β-oxidation-related genes (**b**) in the liver from Vα19 mice (Vα19) and C57BL/6 mice (B6) at 27 weeks of age fed normal chow (NC) and a high-fat diet (HFD). Data on each mRNA were normalized according to the 36B4 mRNA level. The results are shown as the mean ± SD for 4 to 6 mice per group. * *p* < 0.05, ns, not statistically significant.

**Figure 4 ijms-23-15309-f004:**
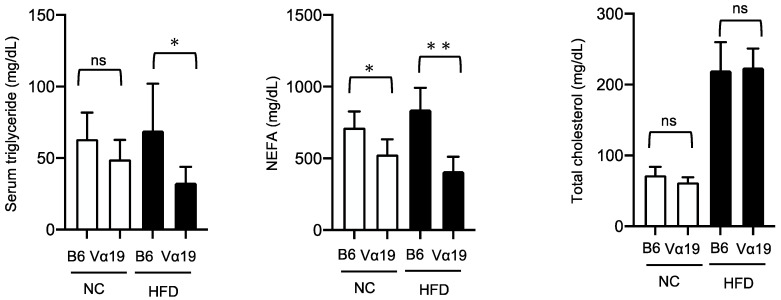
Serum triglyceride and non-esterified fatty acid levels were decreased in Vα19 mice. Serum triglyceride, non-esterified fatty acids (NEFA), and total cholesterol levels in Vα19 mice (Vα19) and C57BL/6 mice (B6) at 27 weeks of age fed normal chow (NC) and a high-fat diet (HFD). The results are shown as the mean ± SD for 7 to 9 mice per group. * *p* < 0.05, ** *p* < 0.005, ns, not statistically significant.

**Figure 5 ijms-23-15309-f005:**
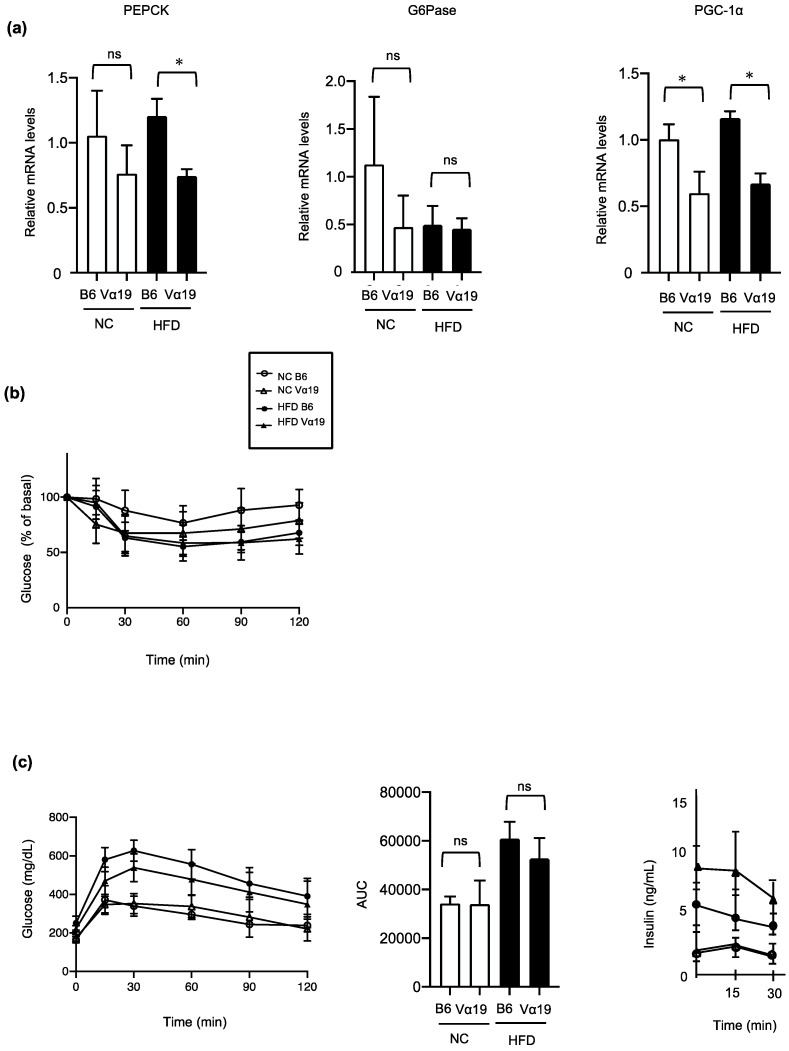
Hepatic gluconeogenesis-related genes decreased, but systemic glucose metabolism and insulin sensitivity were not altered, in Vα19 mice. (**a**) mRNA levels for hepatic gluconeogenesis-related genes in Vα19 mice (Vα19) and C57BL/6 mice (B6) at 27 weeks of age fed normal chow (NC) and a high-fat diet (HFD). Data on each mRNA were normalized according to the 36B4 mRNA level. (**b**,**c**) Intra-peritoneal insulin tolerance test (ipITT) and (**b**) intra-peritoneal glucose tolerance test (ipGTT) (**c**) ipITT and ipGTT were performed after fasting for 4 h. Open circles, B6 mice fed NC; open triangles, Vα19 mice fed NC; closed circles, C57BL/6 mice fed HFD; closed triangles, Vα19 mice fed HFD. * *p* < 0.05, ns, not statistically significant.

**Figure 6 ijms-23-15309-f006:**
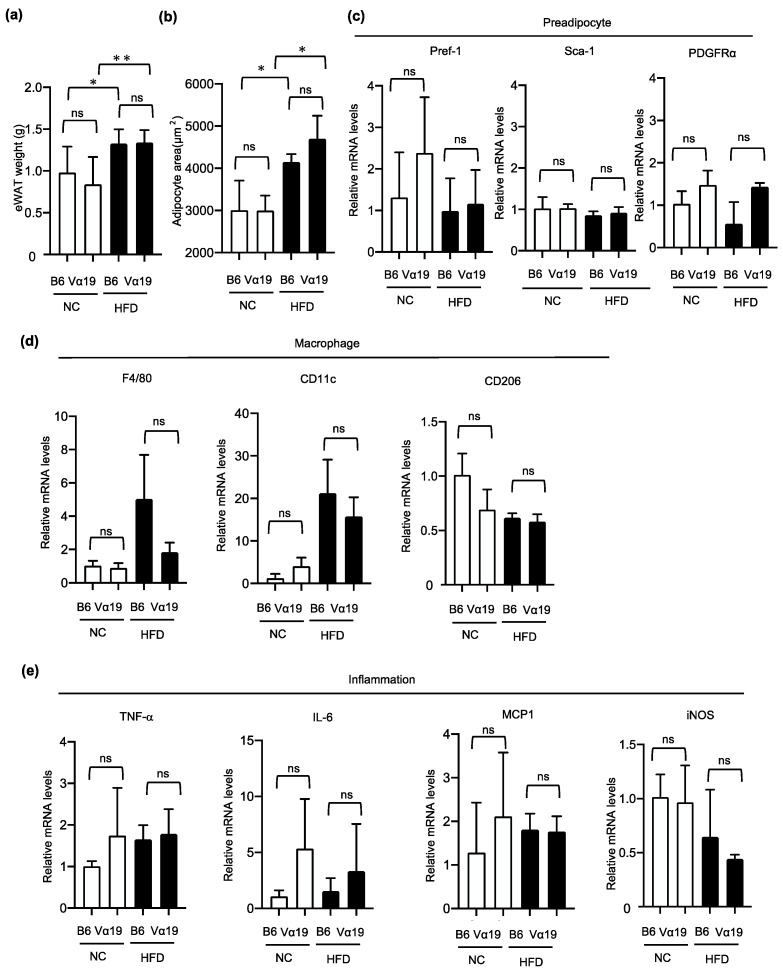
High-fat diet–induced inflammatory remodeling of epididymal adipose tissue was not altered in Vα19 mice. (**a**,**b**) Weight (**a**) and mean adipocyte size (**b**) in epididymal adipose tissue (eWAT) from Vα19 mice (Vα19) and C57BL/6 mice (B6) at 27 weeks of age fed normal chow (NC) and a high-fat diet (HFD). (**c**–**e**) mRNA levels of preadipocyte-related (**c**), macrophage-related (**d**), and inflammation-related (**e**) genes in Vα19 mice and B6 mice. Data on each mRNA were normalized according to the 36B4 mRNA level. * *p* < 0.05, ** *p* < 0.005, ns, not statistically significant.

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
