# Peer review of "Increased Number of Mucosal-Associated Invariant T Cells Is Associated with the Inhibition of Nonalcoholic Fatty Liver Disease in High Fat Diet–Fed Mice"

_ijms, 2022, doi:10.3390/ijms232315309_

Round 1
Reviewer 1 Report
In their manuscript Kishi and co-workers investigated the possible action of mucosal-associated invariant T (MAIT) cells in the pathogenesis of Non-alcoholic fatty liver disease (NAFLD) by using a recently developed mice line named Vα19 mice that are characterized of a high prevalence of MAIT cells. They observed that when feed with a high-fat diet (HFD) V19 mice developed less hepatic steatosis without significant changes of liver inflammatory markers. Moreover, serum triglyceride and non-esterified fatty acids (NEFA) were lower in V19 mice, while glucose tolerance, insulin sensitivity, inflammation in adipose tissues, or intestinal permeability compared with controls. From the data the authors propose that MAIT cells might restrain lipogenesis and liver lipid accumulation during the onset of NAFLD.
At present the role of MAIT cells in the mechanisms leading to the evolution of NAFLD are poorly understood due to the paucity of these cells in mice liver. The experimental approach used by Kishi and co-workers to increase Vα19 MAIT cells in rodents is interesting. However, it is difficult to appreciate whether these mice differ from wild-type in the liver prevalence of other inflammatory/immune cells since these data were not reported.
Beside that the possible involvement of MAIT cells in causing the modest reduction of liver steatosis observed in not completely convincing as, from figure 1, MAIT cell fraction in the liver of Vα19 mice receiving the HFD is about 50% lower than that of mice receiving the normal diet.
It is possible that the lowering in NEFA observed in Vα19 mice either at homeostasis or following HFD might account for the reduction in liver triglyceride accumulation.
Considering that so far the strongest evidence in MAIT cell role in NAFLD relates to their possible contribution in supporting the disease evolution to fibrosis I would suggest to investigate how Vα19 mice behave when fed a high fat carbohydrate diet containing 1.5% cholesterol, also known as Western diet that at difference from the simple HFD causes extensive liver inflammation and significant fibrosis after 16 weeks of treatment. Using this model the authors would be able to dissect MAIT cell role in NAFLD.
Author Response
Response to Reviwer1 Comments
We would like to thank the editor and reviewers for their careful review and valuable suggestions. We agreed with all the critiques and suggestions we received. We would like to respond to each of the suggestions as follows.
Reviewer1
In their manuscript Kishi and co-workers investigated the possible action of mucosal-associated invariant T (MAIT) cells in the pathogenesis of Non-alcoholic fatty liver disease (NAFLD) by using a recently developed mice line named Vα19 mice that are characterized of a high prevalence of MAIT cells. They observed that when feed with a high-fat diet (HFD) Va19 mice developed less hepatic steatosis without significant changes of liver inflammatory markers. Moreover, serum triglyceride and non-esterified fatty acids (NEFA) were lower in Va19 mice, while glucose tolerance, insulin sensitivity, inflammation in adipose tissues, or intestinal permeability compared with controls. From the data the authors propose that MAIT cells might restrain lipogenesis and liver lipid accumulation during the onset of NAFLD.
At present the role of MAIT cells in the mechanisms leading to the evolution of NAFLD are poorly understood due to the paucity of these cells in mice liver. The experimental approach used by Kishi and co-workers to increase Vα19 MAIT cells in rodents is interesting. However, it is difficult to appreciate whether these mice differ from wild-type in the liver prevalence of other inflammatory/ immune cells since these data were not reported.
We are deeply grateful reviewer1 for the review and valuable comments.
Va19 mouse was generated recently, and the phenotype of this novel mouse model has not been fully evaluated, including the changes in inflammatory/immune cells in the liver. As reviewer1 pointed out, we cannot deny the possibility that the reduction in steatosis observed in Va19 mouse was due to the changes in the other inflammatory/immune cells. So, we added the following sentences in discussion.
Va19 mouse was generated recently, and the phenotype of this novel mouse model has not been fully evaluated, including the changes in inflammatory/immune cells in the liver. Thus, it is also possible that the reduction in steatosis of Va19 mouse was due to the changes in the other inflammatory or immune cells. (line258 of draft)
Beside that the possible involvement of MAIT cells in causing the modest reduction of liver steatosis observed in not completely convincing as, from figure 1, MAIT cell fraction in the liver of Vα19 mice receiving the HFD is about 50% lower than that of mice receiving the normal diet.
Thank you very much for the important suggestion. We think that the comparison of MAIT cell number between control B6 mouse and Va19 mouse is more important than the comparison between NC-fed mouse and HFD-fed mouse. However, as reviewer1 pointed out, the Figure1(c) was not convincing in the original manuscript. From the data of multiple experiments of flow cytometry, we are thinking that the number of MAIT cells in the liver was not significantly different between HFD-fed and NC-fed mice. According to the suggestion of reviewer1, in this revision, we replaced Figure1(c) with better one, and the results of multiple experiments was presented in bar graph.
It is possible that the lowering in NEFA observed in Vα19 mice either at homeostasis or following HFD might account for the reduction in liver triglyceride accumulation.
We agree with the suggestion of reviewer1. It is possible that the lowering in NEFA in Vα19 mice might account for the reduced triglyceride accumulation in liver. However, the causal relationship between the lowered NEFA and the reduced hepatic triglyceride accumulation has not been clarified thus far. We added the following sentences in discussion.
In addition, the lowering in NEFA possibly reduced triglyceride accumulation in liver of Vα19 mice. However, the causal relationship between the lowered NEFA and the reduced hepatic triglyceride accumulation has not been clarified. Further experiments are needed to address this possibility. (line291 of draft)
Considering that so far the strongest evidence in MAIT cell role in NAFLD relates to their possible contribution in supporting the disease evolution to fibrosis. I would suggest to investigate how Vα19 mice behave when fed a high fat carbohydrate diet containing 1.5% cholesterol, also known as Western diet that at difference from the simple HFD causes extensive liver inflammation and significant fibrosis after 16 weeks of treatment. Using this model the authors would be able to dissect MAIT cell role in NAFLD.
We are deeply grateful to reviewer1 for a valuable suggestion. As reviewer1 suggested, some recent studies have demonstrated that MAIT cells are involved in the progress of fibrosis in NAFLD (22, 23). We would like to accept the idea and to perform the experiment using Va19 mouse fed on Western diet to address this important issue in the near future. On the other hand, the present study demonstrated a novel role of MAIT cells, that is, MAIT cells are also involved in the process of steatosis, an early step of NAFLD. We added the following sentences in discussion.
In this study, we did not evaluate the fibrotic changes in liver because HFD-feeding did not induce liver fibrosis significantly. To confirm the role of MAIT cells in fibrotic process, we can do the experiments using Va19 mice fed on diet containing not only high fat but also high carbohydrate and cholesterol, which is known to induce fibrosis more strongly than HFD. We would like to address this theme in the next study. (line 246 of draft)

Reviewer 2 Report
The authors investigated the role of MAIT cells in non-alcoholic fatty liver disease (NAFLD) using the novel MAIT cell-enriched mouse model Vaα19, which has a comparable number of MAIT cells to humans. The authors found that increasing MAIT cell numbers reduced hepatic lipid accumulation and disease activity. Although many experiments have been performed to identify the underlying mechanisms, the question of why increased numbers of MAIT cells affect NAFLD remains open.
Major:
· Have the authors considered performing OMICS (transcriptomics or proteomics) of liver and WAT to elucidate the molecular mechanism? Without a mechanistic understanding of the underlying effects, the novelty of this study is limited.
· The authors hypothesise that humoral effects drive lipid synthesis in hepatocytes? Which types are conceivable?
· The authors conclude that MAIT cells could be a therapeutic target in NAFLD. The results presented do not support this conclusion.
· The resolution of the dot blots in Figure 1c needs to be improved. It looks like the TCRβ staining has been truncated. The authors should outline their gating strategy. The results of the flow cytometric analysis should also be presented as bar graphs.
· Poor resolution of HE staining - it is not possible to interpret the images.
· Figure 2c Ballooning error bars missing
Minor:
· In line 56 – 61, references are missing.
· Line 67, reference is missing
· line 70 “other group” reference is missing
· line 99 flow cytometry with space in between
· Figure 2c: x-axis labeling – is there a text field above axis labeling?
· Figure 2e: bad resolution
· Figure 6: X-axis alignment labeling
Author Response
Response to Reviwer2 Comments
Reviewer2
The authors investigated the role of MAIT cells in non-alcoholic fatty liver disease (NAFLD) using the novel MAIT cell-enriched mouse model Vα19, which has a comparable number of MAIT cells to humans. The authors found that increasing MAIT cell numbers reduced hepatic lipid accumulation and disease activity. Although many experiments have been performed to identify the underlying mechanisms, the question of why increased numbers of MAIT cells affect NAFLD remains open.
We are deeply grateful reviewer2 for the review and valuable suggestions.
Major:
- Have the authors considered performing OMICS (transcriptomics or proteomics) of liver and WAT to elucidate the molecular mechanism? Without a mechanistic understanding of the underlying effects, the novelty of this study is limited.
Yes, we also considered that OMICS experiments greatly increase the reliability and value of our current data. Unfortunately, we have not performed OMICS and could not add the data in the revised manuscript because of the short time for this revision. But now we are actually performing the co-culture experiments of hepatocytes and MAIT cells, and are planning any OMICS analyses of culture media to detect any humoral factors, which may explain the mechanisms of how MAIT cells inhibit lipid synthesis. We added the following sentence in discussion of the revised manuscript.
By performing some OMICS analyses of culture media, we aim to detect such humoral factors in the next study. (line 291 of draft)
- The authors hypothesize that humoral effects drive lipid synthesis in hepatocytes? Which types are conceivable?
It is known that MAIT cells secrete Th1/Th17 cytokines, both of which possibly inhibit insulin-induced lipid synthesis in the liver. However, as far as we examined, the expression of TNF-a and IL-6 was not altered between control B6 mouse and Va19 mouse both on NC and HFD (Fig. 2f). In addition, we could not detect the expression of IL-17 in the similar real time RT-PCR experiment (data not shown). Thus far, we have not detected the humoral factors, which may reduce lipid synthesis. As mentioned above, we are doing the co-culture experiments of hepatocytes and MAIT cells which were derived from Va19 mice. By performing some OMICS analyses of culture media as suggested by reviewer2, we aim to detect such humoral factors in the next study. We added the following sentence in discussion.
By performing some OMICS analyses of culture media, we aim to detect such humoral factors in the next study. (line 291 of draft)
- The authors conclude that MAIT cells could be a therapeutic target in NAFLD. The results presented do not support this conclusion.
Thank you very much for the great suggestion. We agree with the suggestion of reviewer2. The conclusion that MAIT cells could be a therapeutic target in NAFLD cannot be drawn from the present results. We deleted the parts indicating that ‘these cells may be a novel therapeutic target for NAFLD’ from abstract and discussion sections (line 26, line 260 and line 293 of draft).
- The resolution of the dot blots in Figure 1c needs to be improved. It looks like the TCRβ staining has been truncated. The authors should outline their gating strategy. The results of the flow cytometric analysis should also be presented as bar graphs.
As reviewer2 pointed out, the dot blots in Figure1(c) could not be seen clearly. The original dot blots were clearer than the printed ones (please see Suppl. Figure2). We replaced the Figure1c with better one in which dot blots could be seen more clearly. In addition, we presented the results of multiple analyses as bar graph in the revised manuscript as Figure 1(c). The gating strategy was outlined in Suppl. Figure2 of the original manuscript. We added the following sentences in results section of the revised manuscript, too.
MAIT cells in the liver were identified by flow cytometry as CD45-positive/TCRb-positive/MR1-tetramer-positive/Ac 6FP-negative cells (Suppl. Fig. 2). (line 99 of draft).
- Poor resolution of HE staining - it is not possible to interpret the images.
As reviewer2 pointed out, the resolution of HE staining in Figure2(a) was very low. The original picture was clearer than the printed ones. We replaced the Figure2(a) with the pictures with higher resolution.
- Figure 2c Ballooning error bars missing
Ballooning grades of hepatocytes with HE staining are evaluated by the score of zero, one and two according to the criteria of Kleiner et al. Because the ballooning grade was evaluated as score two in the all liver sections of HFD-fed mice, mean±SD was 2±0 in the original Figure. For this revision, we performed the additional analyses for ballooning in almost 20 liver sections, and we found that ballooning grade was estimated as score one in some sections. Results were not significantly different between HFD-fed B6 and Va19 mice even after adding the new data, and the evaluation for ballooning was not different from the original data. We drew the new bar graph from the data including new analyses, and we replaced the bar graph of Figure2(c).
Minor:
- In line 56 – 61, references are missing.
Please find references for line 56-61 are reference [10-13].
- Reantragoon, R., et al., Antigen-loaded MR1 tetramers define T cell receptor heterogeneity in mucosal-associated invariant T cells. J Exp Med, 2013. 210(11): p. 2305-20.
- Tilloy, F., et al., Thymic dependence of invariant V alpha 14+ natural killer-T cell development. Eur J Immunol, 1999. 29(10): p. 3313-8.
- Kjer-Nielsen, L., et al., MR1 presents microbial vitamin B metabolites to MAIT cells. Nature, 2012. 491(7426): p. 717-23.
- Corbett, A.J., et al., T-cell activation by transitory neo-antigens derived from distinct microbial pathways. Nature, 2014. 509(7500): p. 361-5.
- Line 67, reference is missing
Please find references for line 67-68 are reference [20, 21].
- Jeffery, H.C., et al., Biliary epithelium and liver B cells exposed to bacteria activate intrahepatic MAIT cells through MR1. J Hepatol, 2016. 64(5): p. 1118-1127.
- Riva, A., et al., Mucosa-associated invariant T cells link intestinal immunity with antibacterial immune defects in alcoholic liver disease. Gut, 2018. 67(5): p. 918-930.
- line 70 “other group” reference is missing
As reviewer2 pointed out, the reference for ‘another group’ from line 70 was missing. It was reference [30] in the original manuscript. We referred the paper as reference [24] in the revised manuscript. Therefore, reference number 24-30 have shifted by one from the original manuscript.
- Li, Y., et al., Mucosal-Associated Invariant T Cells Improve Nonalcoholic Fatty Liver Disease Through Regulating Macrophage Polarization. Front Immunol, 2018. 9: p. 1994.
- line 99 flow cytometry with space in between
‘flowcytometry’ in line 99 was replaced with ‘flow cytometry’.
- Figure 2c: x-axis labeling – is there a text field above axis labeling?
- Figure 6: X-axis alignment labeling
X-axis labelling for Figure2(c) and Figure6 was aligned again, and the figures were replaced with the new ones.
- Figure 2e: bad resolution
Figure2 (e) was replaced with a new one with better resolution.

Round 2
Reviewer 1 Report
The authors have addressed some of the concerns and the text has been modified accordingly. Panel c of the figure 1 has been change in an attempt of showing that following HF diet the Valpha19 mice have a similar propostion of MAIT cells than controls. However, the flow cytometry plot show that in this particular case the overal number of lymphocytes is less than in Valpha 19 mice receiving normal diet. I still feel that this bias might affect the interpretation of the results.
Author Response
Panel c of the figure 1 has been change in an attempt of showing that following HF diet the Valpha19 mice have a similar propostion of MAIT cells than controls. However, the flow cytometry plot show that in this particular case the overal number of lymphocytes is less than in Valpha 19 mice receiving normal diet. I still feel that this bias might affect the interpretation of the results.
We are very grateful to the reviewer1 for evaluating the value of our paper. As reviewer1 pointed out, some readers may find that the representative flow cytometry plots in Fig1c are biased, which may affect the interpretation of the results. I apologize for not being able to provide data that can satisfy all the readers. A summary of the repeated experiments was shown as a bar graph in Fig1c. We believe that the results are true. I would be very grateful if you could accept the data in this state.

Reviewer 2 Report
I thank the authors for addressing all suggested points. The study may be published, but I still think the data is too preliminary and immature.
Author Response
I thank the authors for addressing all suggested points. The study may be published, but I still think the data is too preliminary and immature.
We are very grateful to the reviewer2 for evaluating the value of our paper. As reviewer2 pointed out, this paper still contains some preliminary points. In this revision, following the instructions of the academic editor, we avoided the expression that the increased number of MAIT cells improves NAFLD, and only stated that there is a relationship between the two. In my next research, I would like to further clarify the preliminary points reviewer2 pointed out and the detailed mechanism.
